# The Impact of Obesity on SARS-CoV-2 Pandemic Mortality Risk

**DOI:** 10.3390/nu13103446

**Published:** 2021-09-28

**Authors:** Zhaoping Li, Vijaya Surampudi, David Heber

**Affiliations:** 1Center for Human Nutrition, David Geffen School of Medicine, University of California, Los Angeles, CA 90095, USA; vsurampudi@mednet.ucla.edu (V.S.); dheber@mednet.ucla.edu (D.H.); 2Department of Allergy and Immunology, Veterans Affairs Greater Los Angeles Healthcare System, Los Angeles, CA 90073, USA

**Keywords:** obesity, coronavirus, ACE2, thrombosis, SARS-CoV-2, COVID-19

## Abstract

The COVID-19 pandemic has uncovered the increased susceptibility of individuals with obesity to infection and severe disease leading to hospitalization and death. Studies in New York City demonstrated that after advanced age, obesity was the most common risk factor leading to severe disease and death from COVID-19. While the connection has been recognized, there has not been a general recognition of the potential mechanisms for this link between excess body fat and mortality from this viral pandemic including respiratory complications and sequelae of increased activation of the immune system. Despite plans for vaccination of the global population, the risk community spread of COVID-19 and future pandemics will be linked in part to obesity and immunity. This review will detail a number of potential mechanisms through which obesity may contribute to the lethality of this viral infection. These insights will hopefully lead to a greater emphasis on obesity prevention and treatment as part of the global response to this and future pandemic threats.

## 1. Introduction

COVID-19 was initially perceived as a lung infection due to the entry of the virus into epithelial cells in the lung leading to viral replication and associated inflammation with the characteristic pneumonia of COVID-19. Empirical observations on the association of obesity and diabetes with the incidence of serious COVID-19 infections, hospitalization and mortality have suggested a complex systemic disease due to inflammation that involves immune aspects characteristic of obesity and diabetes pathophysiology. Understanding how obesity and diabetes can set the stage for severe COVID-19 infections and how efforts at prevention and treatment of obesity and type 2 diabetes can help to reduce the risk of populations will add to the armamentarium of public health strategies against this virus and future pandemics. 

COVID 19, also known as SARS-CoV2 is a member of a large family of RNA viruses called coronaviruses causing illnesses ranging from mild self-limited diseases, such as the common cold to previous pandemics of severe including severe acute respiratory syndrome (SARS) infection and the Middle East respiratory syndrome (MERS). These RNA viruses typically move from animal to human hosts and then can lead to human to human transmission. In the last two decades as humans have increasingly encroached on animal habitats in the wild, the risk of pandemics has increased culminating in the COVID-19 pandemic [1]. 

In March 2020, the World Health Organization (WHO) declared the COVID-19 outbreak a global pandemic and issued guidelines to protect health care workers from infection by airborne droplets and contact with infected surfaces. The guidance on prevention, detection and treatment relied on what was known at the time. A report among 4103 patients with COVID-19 disease in New York City found that that body mass index (BMI), which is an estimate of body fat based on weight divided by height squared was the second strongest independent predictor of hospitalization, after old age; a BMI > 40 kg/m2 is characteristic of severe obesity [2]. Moreover, a number of obesity-associated systemic health conditions were also associated with COVID-19 infection and mortality [3]. Among those conditions associated with the need for hospitalization were hypertension, diabetes mellitus, chronic lung disease, and cardiovascular disease, which have in common an association with obesity and nutrition. While there is an association of obesity and obesity-associated conditions with outcomes in COVID-19 infections, this review will examine the potential roles of obesity and nutrition-related mechanisms in the pathogenesis of COVID-19 infections. 

## 2. Obesity-Associated Respiratory Abnormalities

A study of the clinical characteristics of 393 patients in New York City with COVID-19 found that obesity was a more prevalent co-existent condition among patients requiring mechanical ventilation than either diabetes or cardiovascular disease [4]. The clinical manifestations of the disease at presentation in New York were generally similar to those in a large case series from China [5]. However, mechanical ventilation was employed ten times more frequently in New York than in China [4]. Respiratory failure was more common among the subgroup of obese patients, comprising 35.8% of the patients studied in New York. These observations clearly implicate obesity as a risk factor for mechanical ventilation if that is the primary thrust of hospital therapy of COVID-19 when viewed as primarily a respiratory event. 

Obesity is associated with a number of physiological limitations of respiratory function including reduction in lung compliance, reduction in lung volumes, increases in airway resistance, reduction in respiratory muscle strength, and ventilation/perfusion mismatches [6]. In patients with increased abdominal obesity, pulmonary function is further compromised when supine in bed due to restricted diaphragmatic mobility and rib movement, making ventilation more difficult [7]. In a study of patients with acute respiratory distress syndrome (ARDS), a prone position was found to improve oxygenation in obese but not lean patients by comparison to a supine position [8]. As a result of these insights and pragmatic experiences in intensive care units, it became standard procedure to place COVID-19 patients requiring mechanical ventilation in the face-down prone position to improve oxygenation [9]. 

The original approach to treating patients hospitalized with COVID-19 emphasized the treatment of ARDS with mechanical ventilation. A retrospective study of 99 patients from Wuhan China found ARDS in 17% of patients and 11% worsened in a short period of time and died of multiple organ failure [9]. In studies of other respiratory infections, including severe or fatal H1N1 influenza pneumonia, rates of severe and morbid obesity (BMI > 35 kg/m^2^ and > 40 kg/m^2^, respectively) were higher than the general population [10]. 

Understanding of the pathophysiology of ARDS has evolved from emphasizing respiratory mechanics [11] to considering the impact on immune function. Laboratory findings in COVID-19 patients with ARDS included hypoalbuminemia, increased lactate dehydrogenase, and evidence of immune dysfunction including decreased lymphocyte and neutrophil populations, elevated C-reactive protein, and decreased CD8 count [12]. Furthermore, a cytokine storm syndrome has been identified as a fatal event in which there is a flood of pro-inflammatory mediators leading to systemic organ failure and death [13]. 

## 3. Human Angiotensin-Converting Enzyme 2 (ACE2)

The involvement of systemic factors in COVID-19 pathophysiology begins with the initial infection and entry point to cells in the lung epithelium via the Angiotensin converting enzyme-2 (ACE2) membrane-bound carboxypeptidase. ACE2 is found predominantly in the cells of the bronchus and the lung. However, it is also expressed on the membranes of cells in the heart, blood vessels, kidneys, duodenum, and small intestine. ACE 2 is a negative regulator of the renin-angiotensin system and a facilitator of amino acid transport in addition to its role in lung infection by SARS-CoV and SARS-CoV-2 [14]. The COVID-19 spike protein enters human cells by binding to a specific binding site on ACE2 protein (see Figure 1). 

An increase in the expression of ACE-2 in obesity may have a role in the spread of the virus and in the symptomatology of affected patients. In addition, the loss of ACE2 function following binding by COVID-19 spike protein could compromise the essential functions of ACE2 in the Renin-Angiotensin Axis which is to inactivate angiotensin II which promotes hypertension. It has been demonstrated that the coronavirus spike protein binds to ACE2, leading to ACE2 down-regulation and a resulting increase in the production of angiotensin II by ACE [15,16]. 

One group from China reported that ACE2 expression in adipose tissue was greater than in lung tissue. This observation suggests that adipose tissue may be able to promote the spread of COVID-19. While the amounts of ACE2 expressed by adipocytes were similar between non-obese and obese individuals, obese individuals have more adipose tissue thus increasing the number of ACE2 expressing cells in the body. 

In prior studies of H1N1 influenza, symptomatic obese adults were shown to shed influenza A virus in amounts 42% greater than non-obese adults. This suggests obesity may play an important role in influenza transmission and possibly in the transmission of COVID19 [17].

While some clinicians originally raised the possibility that ACE inhibitors (ACEI) or angiotensin receptor blockers (ARBs), used as a treatment for hypertension in COVID-19 patients could increase the severity and mortality of COVID-19 based on the fact that ACE inhibitors and ARBs increase ACE 2 expression [18], there is no evidence supporting worse outcomes with the use of these drugs. In fact, a retrospective, multi-center study of 1128 adult patients reported that among hospitalized COVID-19 patients with hypertension, inpatient use of ACEI/ARB was actually associated with a lower risk of all-cause mortality compared with hypertensive COVID-19 patients not taking ACEI/ARB [19]. Chronic ACE inhibition has been observed to improve endothelial dysfunction which may explain the benefit observed [20]. Another study of 6272 COVID-19 patients and 30,759 matched controls, found no evidence that ACE inhibitors or ARBs affected the risk of COVID-19 [21]. 

## 4. Hypercoagulation and Thrombosis

Several reports and autopsies have reported microvascular thrombi in cardiac and renal tissue in fatal COVID-19 cases [22]. In 388 consecutive symptomatic patients with laboratory-proven COVID-19 admitted to a university hospital in Milan, Italy, thromboembolic events occurred at a cumulative rate of 21% with half diagnosed within 24 h of admission [23]. This finding is especially concerning following the report of five cases of large-vessel stroke in patients younger than 50 years of age in New York City [24]. 

It is being reported that significantly elevated D-dimer may be associated with fatal outcomes in people with COVID-19. While this is possibly related to systemic pro-inflammatory cytokine responses, there are clinical observations that some patients have increasing D-dimer levels after therapy suggesting a possible role of COVID-19 in the development of thrombosis leading to fatalities [15]. 

It is thought that obesity, which is a low-grade chronic inflammatory state, is linked to an enhanced platelet response, mild-to-moderate hypercoagulability, and reduced fibrinolysis [25]. SARS-COV-2 may be exacerbating the underlying chronic inflammation and higher risk of thrombosis in patients who are obese leading to a potential increase in thrombosis contributing to morbidity and mortality. There is very little high-quality evidence supporting specific approaches for the prevention or treatment of coagulation disorders associated with COVID-19. Both standard thromboprophylaxis and treatment of deep venous thrombosis are being carried out, but there is a clear need for more research to understand the coagulation disorders associated with COVID-19 which are distinct from disseminated intravascular coagulation (DIC) associated with other inflammatory disorders [1,26].

## 5. Type 2 Diabetes Mellitus and Cardiovascular Disease 

Obesity is a risk factor for the development of type 2 diabetes and cardiovascular diseases including hypertension. In Chinese studies of COVID-19 patients, hypertension was present in nearly 21%, type 2 diabetes mellitus in nearly 11%, and established cardiovascular disease (CVD) in approximately 7% of patients [27]. 

In a study of 1591 Italian COVID-19 patients, 68% had at least 1 obesity-associated comorbidity and 49% had hypertension. In a study of 5700 COVID-19 patients in New York City, 56.6% had hypertension and 33.8% had type 2 diabetes mellitus while 41.7% were obese [11]. 

In addition to the above co-morbidities associated with obesity, the low-grade metabolic inflammation associated with obesity can impair immune function and the body’s ability to fight against a COVID-19 infection promoting community spread [28]. 

## 6. Vitamin D Deficiency

COVID-19 outbreaks worsen in the winter when 25-hydroxyvitamin D (25(OH)D) levels are reduced at northern latitudes. In countries below the 35 degrees North latitude across the southern hemisphere there are lower Covid-19 mortality rates compared to countries further north [29].

Obese people are known to have lower serum 25 (OH)D levels as this fat-soluble hormone partitions into body fat. Serum 25(OH)D is approximately 20% lower in obese people than in normal weight individuals [30,31]. Moreover, reduced 25(OH)D levels are observed 40% to 80% of obese individuals in survey studies. Free 25(OH)D and 1,25 dihydroxy vitamin D are also observed to be lower in obesity [31].

Studies including epidemiological investigations and clinical trials provide evidence that vitamin D deficiency may lead to an increased risk of influenza and other respiratory tract infections. Low levels of Vitamin D have also been observed in HIV-infected individuals. Cell culture experiments support the concept that vitamin D exerts direct anti-viral effects, particularly against enveloped viruses. Though the mechanism of action of vitamin D against viral infection has not been established, active hypotheses include inducing anti-inflammatory cytokines, inducing cathelicidins and defensins that can lower viral replication rates, and reducing pro-inflammatory cytokines which can injure the lung epithelium leading to diffuse pneumonia [32].

## 7. Obesity, Age, and Gender

Advanced age leads to greater susceptibility to respiratory and systemic infectious diseases including COVID-19. Infection in the elderly is also associated with increased mortality due to reduced immune function with aging [33,34]. 

Sarcopenic obesity with loss of muscle and gain of fat is common in postmenopausal women and older men [35]. Increased abdominal fat is common in both genders with age, but men suffer a higher mortality rate from COVID-19 than women [5,36]. In a retrospective case study from the Lombardy Region of Italy in 1591 consecutive patients with laboratory-confirmed COVID-19 referred for ICU admission, 1304 (82%) were male [37]. While women have both lower and upper body fat, men exclusively gain abdominal fat except under unusual circumstances [38,39]. Upper body fat in men is associated with metabolic abnormalities including inflammation, diabetes and hypertension known to be associated with increased risk of infection and mortality from COVID-19. Therefore, the higher incidence of severe infection and mortality in men may be ascribed to excess intra-abdominal fat and associated immune dysfunction.

To date, autopsy examinations demonstrate many of the histopathological features in the lung similar to those resulting from other coronavirus infections such SARS and MERS, but studies also document additional damage beyond the lungs including endothelial dysfunction with inflammatory cell infiltration. These effects of the infection could be implicated in the multiorgan dysfunction observed in severely ill patients. The ACE2 receptors to which COVID-19 spike proteins bind on lung epithelial cells are also found in the endothelium of many organs. While changes in the immune system, heart and muscle are known to occur in the elderly, there is not enough data available from autopsy examinations demonstrating the multiorgan pathophysiology of severe COVID-19 infections. Most autopsy studies focus on the lungs with only a few studies examining other organs and systemic changes.

Despite the multifaceted and complex process of aging, obesity and gender have emerged as major risk factors during the COVID-19 pandemic. Furthermore, the prevalence of obesity in combination with sarcopenia (the age-related loss of muscle mass and strength or physical function) is increasing in adults aged 65 years and older. A number of these patients can be metabolically impaired at normal body weight, increasing their risk for infection [40]. This is not a rare association, as a large proportion of adults over the age of 65 can be identified as having sarcopenic obesity [35].

## 8. Racial Disparity 

Currently New York City and Chicago are reporting a disproportionately high number of severe COVID-19 cases in the African-American community. In Chicago, 53% of the city’s deaths are in African-Americans, despite African-Americans making up just 30% of the city’s population [41]. Obesity affects some groups more than others with non-Hispanic blacks having had the highest age-adjusted prevalence of obesity (49.6%) [42]. In Chicago, the obesity rate for non-Hispanic blacks is 39.3% vs. 23.7% for non-Hispanic whites [43]. Obesity was recognized as an independent risk factor for influenza during the 2009 H1N1 influenza pandemic while significant race/ethnicity-related disparities in potential risk from H1N1 influenza suggested the potential role of obesity in the observed racial disparity [44,45]. 

## 9. Nutritional Intervention in COVID-19

Prior experience demonstrating the impact of obesity on mortality from H1N1 Influenza provides support for physicians and allied health personnel to emphasize nutrition and obesity treatment as part of the overall care for patients with obesity and COVID-19. 

The United Kingdom, Italy and China have started to evaluate outcome data in relation to BMI of hospitalized patients from SARS-CoV-2. Until more research is able to be done clinicians need to increase vigilance especially in the early stages for priority on detection/testing, and aggressive therapy for patients with obesity and SARS-COV-2 infections. Moreover, obesity affects immune function in complex ways and obesity, diabetes, and hypertension are emerging as risk factors for poor outcome after COVID-19 infection. The COVID-19 pandemic is an early warning to address the fundamental challenges of the global obesity epidemic in view of impacts on transmission and mortality globally for the COVID-19 pandemic and into the future for other pandemic viral infections. 

The interaction of nutrition and immunity is well-established and loss of immune function with protein-energy malnutrition during starvation has been documented by numerous studies to lead to death with depletion of the body cell mass [46]. Other nutritional deficiencies of micronutrients have also been demonstrated to impair immune function [47]. Emerging science indicates that for other nutrients including omega-3 polyunsaturated fatty acids, zinc, iron, selenium, and vitamin E, there is a potential benefit to supplementing above the current recommended dietary allowances to ensure adequate support of immune function and resistance to infection [47]. In addition to well-known micronutrients, there are a large number of phytonutrients from fruits, vegetables, and spices, which have the potential to support immune function [48]. The discovery of the microbiome and the fact that 70% of the immune system is associated with the gastrointestinal tract has revolutionized our understanding of the relationship of nutrition and immunity [49]. For example, it is known that the diversity of the microbiome decreases with aging, which may help to explain in part the increased risk associated with COVID-19 in the elderly. Furthermore, changes in the microflora of the gut have been observed in patients infected with COVID-19. These different areas of nutritional research on COVID-19 and immunity are still at early stages, but common-sense nutrition and supplementation provides a low-cost strategy for public health nutritional approaches to the COVID-19 pandemic. 

Obesity is associated with numerous age-related chronic diseases and increased mortality including most recently, COVID-19 deaths as indicated in this review. While bodyweight is used to document the incidence of obesity in many countries around the world using the Body Mass Index (weight divided by height squared), it is increased intra-abdominal body fat and endocrine and metabolic dysfunction that is most likely to contribute to the systemic inflammation characteristic of severe COVID-19 infections. Obesity can be addressed with a combination of dietary interventions and exercise in individuals and with significant changes in the food supply, which can be supported through government policies that increase the intake of nutrient-rich foods, such as fruits, vegetables and lean proteins while reducing the availability of sweetened beverages, refined carbohydrates, and high fat/high sugar snack foods. While the primary focus of COVID-19 prevention has been testing for infection and vaccination, in preparing for future pandemics more attention should be given to the prevention and treatment of obesity and obesity-associated diseases including type 2 diabetes mellitus.

## Figures and Tables

**Figure 1 nutrients-13-03446-f001:**
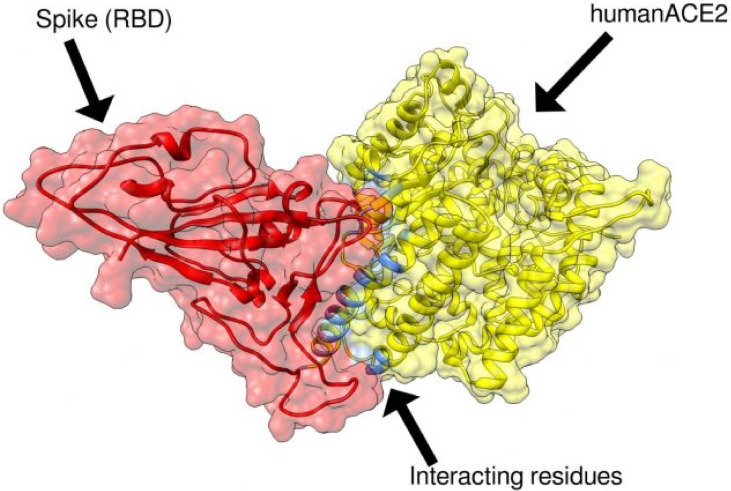
The COVID-19 spike protein shown in red binds to the human ACE2 receptor shown in yellow at known amino acid residues prior to gaining entry into epithelial cells lining the lung. Vaccines are designed to increase human antibodies capable of binding to the COVID-19 spike protein thereby blocking the virus from binding to ACE2 receptor and preventing infection [credit: Flinders University].

## Data Availability

Not applicable.

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
