# Peer review of "The Impact of Obesity on SARS-CoV-2 Pandemic Mortality Risk"

_nutrients, 2021, doi:10.3390/nu13103446_

Round 1

Reviewer 1 Report

The authors have revised the manuscript completely and appropriately, I suggest to publish this paper without further revision.

Author Response

Nutrients - 1395678

Dear Reviewers,

Thank you for giving us the opportunity to submit a revised draft of the manuscript. We appreciate the time and effort that all the reviewers dedicated to providing feedback on our manuscript and are grateful for the insightful comments on and valuable improvements to our paper. We have tried to address the suggestions of the reviewers. Those changes are highlighted in the manuscript. Please see below, in blue, for a point-by-point response to the reviewers’ comments and concerns. The manuscript file has tracked changes.

Reviewers Comments to The Authors:

Reviewer 1: None

Reviewer 2: The topic of this review is very much welcomed, the review is well structured, but seems very short. Naturally, there aren't much data available yet to review ever since the fairly recent outbreak of the COVID-19 pandemic.  There are some minor issues that need to be fixed to improve readability and assure a broader readership, although, in general, the English quality is good.

  1. Words missing, spelling, minor English language issues: in the one-sentence summary "of" should be replaced with "that". In the introduction, "pandemics of severe ... including" - something is missing between "severe" and "including". At the bottom of the first page "that that" should be fixed (the word is doubled). In the yellow highlighted paragraph on page 5, "IN addition" or, alternatively, "additionAL".
  2. There should be a comma before "which" in most instances throughout the manuscript.
  3. In the second paragraph of Section 4, there is a citation for the NYC study, but no citation for the Italian study (probably citation [37] from the reference list in needed here)
  4. Some acronyms are not detailed anywhere in the whole manuscript, e.g. ARDS (although well-known jargon in the medical community, for the inexperienced reader a list of abbreviations for the entire paper would help a lot). Also, BMI is another acronym not explained until the very last paragraph of the manuscript, although it appears three times before that paragraph. 

Thank you for the detailed review. We have reread the manuscript have addressed the changes above and the changes have been tracked in the manuscript.

Reviewer 2 Report

Dear Authors,

The topic of this review is very much welcomed, the review is well structured, but seems very short. Naturally, there aren't much data available yet to review ever since the fairly recent outbreak of the COVID-19 pandemic.  There are some minor issues that need to be fixed to improve readability and assure a broader readership, although, in general, the English quality is good.

  1. Words missing, spelling, minor English language issues: in the one-sentence summary "of" should be replaced with "that". In the introduction, "pandemics of severe ... including" - something is missing between "severe" and "including". At the bottom of the first page "that that" should be fixed (the word is doubled). In the yellow highlighted paragraph on page 5, "IN addition" or, alternatively, "additionAL".
  2. There should be a comma before "which" in most instances throughout the manuscript.
  3. In the second paragraph of Section 4, there is a citation for the NYC study, but no citation for the Italian study (probably citation [37] from the reference list in needed here)
  4. Some acronyms are not detailed anywhere in the whole manuscript, e.g. ARDS (although well-known jargon in the medical community, for the inexperienced reader a list of abbreviations for the entire paper would help a lot). Also, BMI is another acronym not explained until the very last paragraph of the manuscript, although it appears three times before that paragraph. 

Author Response

Nutrients - 1395678

Dear Reviewers,

Thank you for giving us the opportunity to submit a revised draft of the manuscript. We appreciate the time and effort that all the reviewers dedicated to providing feedback on our manuscript and are grateful for the insightful comments on and valuable improvements to our paper. We have tried to address the suggestions of the reviewers. Those changes are highlighted in the manuscript. Please see below, in blue, for a point-by-point response to the reviewers’ comments and concerns. The manuscript file has tracked changes.

Reviewers Comments to The Authors:

Reviewer 1: None

Reviewer 2: The topic of this review is very much welcomed, the review is well structured, but seems very short. Naturally, there aren't much data available yet to review ever since the fairly recent outbreak of the COVID-19 pandemic.  There are some minor issues that need to be fixed to improve readability and assure a broader readership, although, in general, the English quality is good.

  1. Words missing, spelling, minor English language issues: in the one-sentence summary "of" should be replaced with "that". In the introduction, "pandemics of severe ... including" - something is missing between "severe" and "including". At the bottom of the first page "that that" should be fixed (the word is doubled). In the yellow highlighted paragraph on page 5, "IN addition" or, alternatively, "additionAL".
  2. There should be a comma before "which" in most instances throughout the manuscript.
  3. In the second paragraph of Section 4, there is a citation for the NYC study, but no citation for the Italian study (probably citation [37] from the reference list in needed here)
  4. Some acronyms are not detailed anywhere in the whole manuscript, e.g. ARDS (although well-known jargon in the medical community, for the inexperienced reader a list of abbreviations for the entire paper would help a lot). Also, BMI is another acronym not explained until the very last paragraph of the manuscript, although it appears three times before that paragraph. 

Thank you for the detailed review. We have reread the manuscript have addressed the changes above and the changes have been tracked in the manuscript.

This manuscript is a resubmission of an earlier submission. The following is a list of the peer review reports and author responses from that submission.

Round 1

Reviewer 1 Report

Dear Authors
The review entitled "The Impact of Obesity on SARS-CoV-2 Pandemic Mortality Risk" is very interesting and reflects the authors' knowledge of the subject. 
However, the acquisition of knowledge that the articles produce on the authors should not be confused with the generation of knowledge. A review, whether systematic or scoping, generates knowledge and has a methodology and this is not reflected in the attached document.
At least, the manuscript lacks an explanation of the search equations of the bibliography, the methodology to establish the equations, the databases, (PubMed, Scopus, WOS, Cinhal, ...)  where the searches have been performed, whether individual or peer review has been done, the studies discarded and why they were discarded, and an assessment of the quality of these studies. A section on limitations should be included since the review probably only shows the top of the iceberg. 
In any case, if it is a Narrative Review (as it seems to be), remind the authors that this type of review is at the bottom of the pyramid, exposed to the possibility of presenting a high risk of bias, mainly due to its subjectivity and lack of methodology.
I encourage the authors to face the challenge of conducting the review following the usual procedures of a Systematic Review or Scoping Review. I am sure they will find more studies and make sure that they have left almost none to review. 

Kind regards

Author Response

This manuscript entitled "The Impact of Obesity on SARS-CoV-2 Pandemic Mortality Risk"  is indeed a Narrative Review. It is meant to heighten awareness of the association of obesity and COVID-19 and not to prove a connection. It leaves open the need for further research and does not aim to present a biased viewpoint. We do not wish to write a systematic review of the topic.

Reviewer 2 Report

The subject matter of the article is very interesting as it represents a topical issue. The introduction is very confusing and superficial: the structure of SARS-COV2 and what mechanism is known to date about its penetration of the organism should be better described in order to give the reader a general idea about SARS-COV2 and the infection COVID19.

In the paragraph where the authors talk about thrombosis, they could implement the bibliography with this interesting article:  Zanza C, Racca F, Longhitano Y, Piccioni A, Franceschi F, Artico M, Abenavoli L, Maiese A, Passaro G, Volonnino G, La Russa R. Risk Management and Treatment of Coagulation Disorders Related to COVID-19 Infection. Int J Environ Res Public Health. 2021 Jan 31;18(3):1268. doi: 10.3390/ijerph18031268.

In the paragraph Obesity, Age, and Gender, in order to make the study more interesting, the authors should also discuss the incidence of obesity found in COVID19 deaths during autopsies as well written in this article: Autopsy findings in COVID-19-related deaths: a literature review Aniello Maiese, Alice Chiara Manetti, Raffaele La Russa,Marco Di Paolo,Emanuela Turillazzi,Paola Frati,and Vittorio Fineschi

Author Response

Comment: The introduction is very confusing and superficial: the structure of SARS-COV2 and what mechanism is known to date about its penetration of the organism should be better described in order to give the reader a general idea about SARS-COV2 and the infection COVID19.

We have re-written the introduction as follows:

While COVID-19 was initially perceived as a lung infection due to the point of entry of the virus and the characteristic pneumonia of COVID-19, empirical observations on  the association of obesity and diabetes with the incidence of serious infections, hospi-talization and mortality have suggested a complex systemic disease due to inflammation that involves immune aspects characteristic of obesity and diabetes pathophysiology. Understanding how obesity and diabetes can set the stage for severe COVID-19 infections and how efforts at prevention and treatment of obesity and type 2 diabetes can help to reduce the risk of populations will add to the armamentarium of public health strategies against this virus and future pandemics.

Comment: In the paragraph where the authors talk about thrombosis, they could implement the bibliography with this interesting article:  Zanza C, Racca F, Longhitano Y, Piccioni A, Franceschi F, Artico M, Abenavoli L, Maiese A, Passaro G, Volonnino G, La Russa R. Risk Management and Treatment of Coagulation Disorders Related to COVID-19 Infection. Int J Environ Res Public Health. 2021 Jan 31;18(3):1268. doi: 10.3390/ijerph18031268.

We have added the following referencing the above paper:

There is very little high-quality evidence supporting the approaches for the prevention or treatment of the coagulation disorders associated with COVID-19. Both standard thromboprophylaxis and treatment of deep venous thrombosis is being carried out, but there is a clear need for more research to understand the coagulation disorders associated with COVID-19 which are distinct from Disseminated Intravascular Coagulation (DIC) associated with other inflammatory disorders (A1).

A1. Franceschi F, Artico M, Abenavoli L, Maiese A, Passaro G, Volonnino G, La Russa R. Risk Management and Treatment of Coagulation Disorders Related to COVID-19 Infec-tion. Int J Environ Res Public Health. 2021 Jan 31;18(3):1268. doi: 10.3390/ijerph18031268. PMID: 33572570; PMCID: PMC7908596.

Comment: In the paragraph Obesity, Age, and Gender, in order to make the study more interesting, the authors should also discuss the incidence of obesity found in COVID19 deaths during autopsies as well written in this article: Autopsy findings in COVID-19-related deaths: a literature review Aniello Maiese, Alice Chiara Manetti, Raffaele La Russa,Marco Di Paolo,Emanuela Turillazzi,Paola Frati,and Vittorio Fineschi

We have added the following referencing also the above paper:

To date, autopsy examinations demonstrate many of the histopathological features in the lung similar to other coronavirus infections such SARS and MERS, but also document addition damage beyond the lungs including endothelial dysfunction with inflammatory cell infiltration. These effects of the infection could be implicated in the multiorgan dysfunction observed in severely ill patients. The ACE2 receptors to which COVID-19 spike proteins bind on lung epithelial cells are also found in the endothelium of many organs. While changes in the immune system, heart and muscle are known to occur in the elderly, there is not enough data available from autopsy examinations ena-bling an understanding of the multiorgan pathophysiology of severe COVID-19 infec-tion. Most autopsy studies focus on the lungs with only a few studies examining other organs and systemic changes (A2).

A2. Maiese A, Manetti AC, La Russa R, Di Paolo M, Turillazzi E, Frati P, Fineschi V. Autopsy findings in COVID-19-related deaths: a literature review. Forensic Sci Med Pathol. 2021 Jun;17(2):279-296. doi: 10.1007/s12024-020-00310-8. Epub 2020 Oct 7. PMID: 33026628; PMCID: PMC7538370. 

Reviewer 3 Report

The authors overview eight parts to summarize the most recent evidence on the role of obesity as a potential risk factor for leading to severe disease and death from COVID-19, including Obesity-Associated Respiratory Abnormalities, Human angiotensin-converting enzyme 2 (ACE2), Hypercoagulation and Thrombosis, Type 2 Diabetes Mellitus and Cardiovascular Disease, Vitamin D Deficiency, Obesity, Age, and Gender, Racial Disparity, and Nutritional Intervention in COVID-19. Overall, I think this manuscript are including: 1.Comprehensive systematic search literature, 2. Complete literature review and 3. Good logical to illustrate the topics

My comments are below:

  1. Although the authors detailed a number of potential mechanisms through which obesity may contribute to the lethality of this viral infection. However, some sections are too simple. For example, “4. Type 2 Diabetes Mellitus and Cardiovascular Disease, 6. Obesity, Age, and Gender, 7. Racial Disparity and 8. Nutritional Intervention in COVID-19”.

  1. The author said “These insights will hopefully lead to a greater emphasis on obesity prevention and treatment as part of the global response to this and future pandemic threats. ”So, I suggest the authors may discuss the strategies on obesity prevention and treatment as part to enrich the review article content.

  1. In Figure 1 legend. The COVID-19 spike protein shown in red binds to the human “ACE2 receptor” shown in yellow ….., please check “ACE2 receptor” or “ACE” ?

  1. On page 3 the last paragraph, There has been concern that ACE inhibitors or angiotensin receptor blockers (ARBs),usually used as treatment for hypertension, could increase the risk of infection and COVID-19 severity. ACE inhibitors (ACEI) and ARBs” increase” ACE 2 expression [18], please check ”please check” or “decrease” ?

Author Response

  1. Although the authors detailed a number of potential mechanisms through which obesity may contribute to the lethality of this viral infection. However, some sections are too simple. For example, “4. Type 2 Diabetes Mellitus and Cardiovascular Disease, 6. Obesity, Age, and Gender, 7. Racial Disparity and 8. Nutritional Intervention in COVID-19”.

We have attempted to highlight the major associations of age-related chronic diseases which are related to obesity and type 2 diabetes mellitus in these paragraphs. We have not gone into great detail as this was outside the scope of this narrative review.

  1. The author said “These insights will hopefully lead to a greater emphasis on obesity prevention and treatment as part of the global response to this and future pandemic threats.”So, I suggest the authors may discuss the strategies on obesity prevention and treatment as part to enrich the review article content.

We have added the following:

Obesity is associated with numerous age-related chronic diseases and increased mortality including most recently COVID-19 deaths as indicated in this review. While body weight is used to document the incidence of obesity in many countries around the world using the Body Mass Index (weight divided by height squared), it is increased intra-abdominal body fat and endocrine and metabolic dysfunction that is most likely to contribute to the systemic inflammation characteristic of severe COVID-19 infections. Obesity can be addressed with a combination of dietary interventions and exercise in individuals and with significant changes in the food supply which can be supported through government policies that increase the intake of nutrient-rich foods such as fruits, vegetables and lean proteins while reducing the availability of sweetened beverages, re-fined carbohydrates, and high fat/ high sugar snack foods. While the primary focus of COVID-19 prevention has been testing for infection and vaccination, in preparing for future pandemics more attention should be given to the prevention and treatment of obesity and obesity-associated diseases including type 2 diabetes mellitus. 

  1. In Figure 1 legend. The COVID-19 spike protein shown in red binds to the human “ACE2 receptor” shown in yellow ….., please check “ACE2 receptor” or “ACE” ?

It is indeed the ACE2 receptor protein not ACE.

  1. On page 3 the last paragraph, There has been concern that ACE inhibitors or angiotensin receptor blockers (ARBs), usually used as treatment for hypertension, could increase the risk of infection and COVID-19 severity. ACE inhibitors (ACEI) and ARBs” increase” ACE 2 expression [18], please check ”please check” or “decrease” ?

We have clarified this to indicate that the initial concern of worsened severity of COVID-19 was not supported by careful studies. We added the following:

While some clinicians originally raised the possibility that ACE inhibitors (ACEI) or angiotensin receptor blockers (ARBs), used as treatment for hypertension in COVID-19 patients could increase the severity and mortality of COVID-19 based on the fact that ACE inhibitors and ARBs increase ACE 2 expression [18], there is no evidence supporting worse outcomes with the use of these drugs. In fact, a retrospective, multi-center study of 1128 adult patients reported that among hospitalized COVID-19 patients with hyper-tension, inpatient use of ACEI/ARB was actually associated with a lower risk of all-cause mortality compared with hypertensive COVID-19 patients not taking ACEI/ARB [19]. Chronic ACE inhibition has been observed to improve endothelial dysfunction which may explain the benefit observed [20].  Another study of 6272 COVID-19 patients and 30,759 matched controls, found no evidence that ACE inhibitors or ARBs affected the risk of COVID-19 [21].  

Round 2

Reviewer 1 Report

I have no comments

Reviewer 2 Report

It’s ok but the authors dont’describe The strutture os SARS-Cov2 as suggested  

Reviewer 3 Report

The authors answered  "It is indeed the ACE2 receptor protein not ACE.". Why was the yellow "humanACE2" in figure 1 rather than "humanACE2 receptor" ? In addition, In Figure 1 legend. The COVID-19 spike protein shown in red binds to the human "ACE2 receptor" shown in yellow at known amino acid residues prior to gaining entry into epithelial cells lining the lung. Vaccines are designed to increase human antibodies capable of binding to the COVID-19 spike protein thereby blocking the virus from binding to "ACE2" and preventing infection. There are "ACE2 receptor" and "ACE2" in this figure legend.  If the first "ACE2 receptor"  is "ACE2 receptor", why the second "ACE2" is "ACE2"? Should the second "ACE2" be "ACE2 receptor" ?